# Uncovering the Expansin Gene Family in Pomegranate (*Punica granatum* L.): Genomic Identification and Expression Analysis

Xintong Xu [1,2], Yuying Wang [1,2], Xueqing Zhao [1,2] and Zhaohe Yuan [1,2,*]

1 Co-Innovation Center for Sustainable Forestry in Southern China, Nanjing Forestry University, Nanjing 210037, China; xuxintong@njfu.edu.cn (X.X.); wangyuying@njfu.edu.cn (Y.W.); zhaoxq402@163.com (X.Z.)
2 College of Forestry, Nanjing Forestry University, Nanjing 210037, China
* Correspondence: zhyuan88@hotmail.com

**Abstract:** Expansins, which are important components of plant cell walls, act as loosening factors to directly induce turgor-driven cell wall expansion, regulate the growth and development of roots, leaves, fruits, and other plant organs, and function essentially under environmental stresses. In multiple species, many expansin genes (*EXPs*) have been cloned and functionally validated but little is known in pomegranate. In this study, a total of 33 *PgEXPs* were screened from the whole genome data of 'Taishanhong' pomegranate, belonging to the EXPA(25), EXPB(5), EXLA(1), and EXLB(2) subfamilies. Subsequently, the composition and characteristics were analyzed. Members of the same branch shared similar motif compositions and gene structures, implying they had similar biological functions. According to *cis*-acting element analysis, *PgEXPs* contained many light and hormone response elements in promoter regions. Analysis of RNA-seq data and protein interaction network indicated that *PgEXP26* had relatively higher transcription levels in all pomegranate tissues and might be involved in pectin lyase protein synthesis, whilst *PgEXP5* and *PgEXP31* might be involved in the production of enzymes associated with cell wall formation. Quantitative real-time PCR (qRT-PCR) results revealed that *PgEXP* expression levels in fruit peels varied considerably across fruit developmental phases. *PgEXP23* was expressed highly in the later stages of fruit development, suggesting that *PgEXP23* was essential in fruit ripening. On the other hand, the *PgEXP28* expression level was minimal or non-detected. Our work laid a foundation for further investigation into pomegranate expansin gene functions.

**Keywords:** pomegranate; expansin gene family; bioinformatics; expression pattern





## 1. Introduction

In plant cells, the cell wall is an essential and distinct structure. It determines cell shape and size, provides mechanical support and stiffness, and is the cell's first barrier against pathogens [1]. Owing to the importance of cell wall enlargement in plant morphogenesis [2], it is becoming a hot focus to study the mechanism of cell wall extension. Previously, the 'acid growth phenomenon' showed that in an acidic environment, the cell wall can be extended without structural changes, and the extending characteristics can be induced or inhibited in a short time [3]. However, the acid growth hypothesis does not address the biochemical essence of cell wall relaxation, for which expansins provide a possible mechanism of action [4].

Expansin, a broad-spectrum protein, not only relaxes and irreversibly stretches cell walls in an acidic environment, but it also enhances cell extensibility. Its function is to regulate intercellular wall component relaxation and increase cell wall flexibility by breaking hydrogen bonds between cellulose microfibers and hemifibers [5]. In land plants, expansins are important regulators of turgor-driven cell wall expansion [6], while the expansin family was a highly ancient and conserved large gene family [7]. The analysis of

gene structure and amino acid sequence showed that expansin genes were derived from a common ancestor and could be divided into four subfamilies: EXPA, EXPB, EXLA, and EXLB [8]. Moreover, studies found EXPA and EXPB subfamily genes mostly act upon plant cell wall extension and the processes of growth and development [9–11], while there was no proof that EXLA and EXLB were active on the cell wall, they play a major role in controlling plant stomata opening and closing [12,13]. Currently, with the advancement of genome sequencing and analysis technology, the expansin gene family has been comprehensively identified in many plants, including Arabidopsis [14], grape [15], apple [16], cotton [8], kiwi [17], and cannabis [18]. Although their sequence composition, structure –function, and number varied substantially among different species [19], expansin genes widely regulate plant growth, meristem growth, root hair emergence, pollen tube entry into stigma and ovary, fruit ripening, pericarp rupture, and other plant growth and development processes. For example, *ZmEXPB13*, an endosperm, specifically expressed genes that influenced seed germination [20]. In Arabidopsis, partial silencing of *AtEXPA7* led to shorter root hairs, and a point mutation in the rice gene *OsEXPA17* resulted in altered root hair emergence [21,22]. The *GgEXPA1* gene from gladiolus was shown to be highly expressed during the elongation stage of stamen filament cells [23]. Overexpressing the *FaEXP2* gene increased pectin content in the cell walls of transgentic lines while decreasing the expression level of genes encoding cell wall degrading enzymes, resulting in hard fruit or late ripening of fruit [24]. The expression of the expansin gene was significantly decreased in crack-prone longan pericarp, showing the gene plays a crucial role in crack resistance creation [25]. Moreover, expansin genes were involved in salt tolerance and drought resistance [26,27], among other functions.

Pomegranate (*Punica granatum* L.) is a species of economically important trees that is widely planted worldwide and is native to Central Asia, including Iran, Afghanistan, and the Caucasus [28,29]. It is well known for its vivid red skin and juicy seeds. Furthermore, the fruit peel and juice extracts are rich in antioxidants, such as polyphenols, and have been suggested to have positive effects in cardiovascular, tumors, diabetes, and other diseases [30–33]. In recent years, scholars have successively assembled several pomegranate genomes, including 'Taishanhong' [28], 'Dabenzi' [29], and 'Tunisia' [34], and acquired high-quality genome maps, offering an essential molecular biological basis for pomegranate genetic improvement. With advancements in molecular biology, we may not only study specific gene family functions bioinformatically but also analyze genes that regulate plant growth and development as well as environmental stress. What is more, these progresses play important roles in revealing the mechanism of development and stress tolerance [35,36]. The research to date on pomegranate expansins (*PgEXPs*) is still in its early stage. Thus, based on the 'Taishanhong' genome, we used bioinformatic approaches to identify members of the pomegranate expansin gene family and analyzed their physicochemical properties, conserved domain, evolutionary relationship, *cis*-acting element, and tissue organ expression. Our study will lay a foundation for further research on the functions of the expansin gene in pomegranate.

## 2. Materials and Methods

### 2.1. Plant Materials

The pomegranate variety for testing was 'Daqingpitian', and the sampling location was the Chinese Pomegranate Expo Park (34°77′ N, 117°48′ E) with sloppy management level. We selected three healthy, disease-free and uniformly growing adult fruit-bearing trees, and took a mixed sampling method to collect samples. A total of six different developmental periods (the dates were 25 August, 3 September, 12 September, 21 September, 30 September, and 9 October, designated as P1~P6, respectively) fruit samples were collected, finally.

### 2.2. Identification and Physicochemical Properties of PgEXP Family Genes

To identify the pomegranate expansin gene family, firstly, the Hidden Markov Model profile of the 'EXP domains (PF01357 and PF03330) was obtained from the Pfam database

(http://pfam.xfam.org/, accessed on 3 February 2023), The pomegranate genome-wide data ('Taishanhong' ASM286412v1) were downloaded from the National Center for Biotechnology Information (NCBI) official website (http://www.ncbi.nlm.nih.gov/, accessed on 3 February 2023). Then, using HMMsearch, we compared sequences with all two conserved domains (E-value $\leq 10^{-10}$) in the pomegranate protein database to get the amino acid sequences of the originally screened pomegranate EXPs. Lastly, we removed redundant sequences manually by Excel, the rest were screened for conserved domains using the NCBI CDD (http://www.ncbi.nlm.nih.gov/cdd, accessed on 3 February 2023) and SMART (http://smart.embl-heidelberg.de, accessed on 3 February 2023) to exclude candidate sequences with missing conserved domains.

The physicochemical properties, which include the isoelectric point (pI), molecular weight (MW), and instability index of *PgEXP* members, were predicted using ExPaSy-Protparam (http://web.expasy.org/protparam/, accessed on 4 February 2023) [37]. Signal peptides were predicted using SignalP 4.1 (http://www.cbs.dtu.dk/services/Signalp/index.php, accessed on 4 February 2023). Subcellular localization prediction was carried out by using Cell-PLoc 2.0 (http://www.csbio.sjtu.edu.cn/bioinf/Cell-PLoc-2/, accessed on 4 February 2023).

### 2.3. Phylogenetic Analysis

In order to classify PgEXPs based on phylogenic tree, the selected amino acid sequences of EXPs from pomegranate, *Arabidopsis*, grape, kiwi, and jujube were aligned by MUSCLE in MEGA 11 software with default settings [38], and the compared sequences were also trimmed with MEGA 11.A Maximum Likelihood (ML) in IQTree2 V2.1.3 and was then constructed under the best-fitting model with 1000 bootstrap replicates. EvolView (http://www.evolgenius.info/evolview/, accessed on 6 February 2023) was used to enhance the phylogenetic tree online. The phylogenetic position of PgEXPs in relation to reference EXPs was used to group them. We collected and summarized published records on the EXP family from other plant species to compare the phylogenetic groups of the EXPs in pomegranate and other species.

### 2.4. Analysis of Conserved Domains, Gene Structure and Protein Conserved Motif

According to the acquired PgEXP protein sequences and gene sequences, we used MUSCLE to compare them as shown by Jalview software. The online program MEME Suit (http://meme-suit.org/, accessed on 8 February 2023) was used to conduct motif analysis. To identify the exon–intron structure of 33 *PgEXP* genes, their annotation information was taken from pomegranate whole genome gff files and uploaded to the web application GSDS (http://gsds.cbi.pku.edu.cn/, accessed on 8 February 2023). Afterward, TBtools [39] was used to illustrate the phylogenetic tree, conserved motifs, and gene structure of *PgEXPs*.

### 2.5. Analysis of Cis-Acting Elements and Protein Interaction Networks

Promoter sequences (1500-bp upstream from the start codon) were extracted from the genome sequence of *PgEXPs*. Then, the online website PlantCARE (http://bioinformatics.psb.ugent.be/webtools/plantcare/html/, accessed on 8 February 2023) was used to analyze potential *cis*-acting elements, the findings visualized by TBtools software. To examine gene co-expression patterns, protein patterns with reasonably high specificity from the String (http://cn.string-db.org, accessed on 10 February 2023) were employed, and the model plant *Arabidopsis thaliana* was chosen as the species parameter.

### 2.6. RNA-Seq Analysis

To analyze expression patterns of *PgEXPs* in different pomegranate tissues and organs, the published transcriptome data of six pomegranate varieties ('Dabenzi', 'Tunisia', 'Baiyushizi', 'Black127', 'Nana', and 'Wonderful') were downloaded from NCBI (http://www.ncbi.nlm.nih.gov/, accessed on 10 February 2023), including outer seed coat, inner seed coat, pericarp, flower, root, leaf, and mixed samples of roots, leaf, flower, and fruit as

shown in Table 1. Then, the transcriptomic data were calculated and analyzed with Kallisto v0.44.0 software (California, USA) [40], and the resulting values were transformed into $\text{Log}_2(\text{TPM}+1)$ (Table S1) and finally, the expression heatmap was created by TBtools.

**Table 1.** Pomegranate transcript data.

| Accession No. | Cultivar | Sample | ID | Reference |
|---|---|---|---|---|
| SRR5279388 | Dabenzi | Outer seed coat | Dabenzi_OSC | [29] |
| SRR5279391 | Dabenzi | Inner seed coat | Dabenzi_ISC | [29] |
| SRR5279394 | Dabenzi | Pericarp | Dabenzi_pericarp | [29] |
| SRR5279395 | Dabenzi | Flower | Dabenzi_flower | [29] |
| SRR5279396 | Dabenzi | Root | Dabenzi_root | [29] |
| SRR5279397 | Dabenzi | Leaf | Dabenzi_leaf | [29] |
| SRR5446592 | Tunisia | Bisexual flowers (3.0–5.0 mm) | 3–5 mm(B) | [34] |
| SRR5446595 | Tunisia | Bisexual flowers (5.1–13.0 mm) | 5.1–13 mm(B) | [34] |
| SRR5446598 | Tunisia | Bisexual flowers (13.1–25.0 mm) | 13.1–25 mm(B) | [34] |
| SRR5446601 | Tunisia | Functional male flowers (3.0–5.0 mm) | 3–5 mm(F) | [34] |
| SRR5446604 | Tunisia | Functional male flowers (5.1–13.0 mm) | 5.1–13 mm(F) | [34] |
| SRR5446607 | Tunisia | Functional male flowers (13.1–25.0 mm) | 13.1–25 mm(F) | [34] |
| SRR5678820 | Tunisia | Inner seed coat | TNS_ISC | [29] |
| SRR5678819 | Baiyushizi | Inner seed coat | BYSZ_ISC | [29] |
| SRR1054190 | Black127 | Mix of leaves, flowers, fruit and roots | Black127 | [41] |
| SRR1055290 | Nana | Mix of leaves, flowers, fruit and roots | Nana | [41] |
| SRR080723 | Wonderful | Pericarp | Wonderful | [42] |

*2.7. RNA Isolation, Reverse Transcription and Quantitative Real-Time PCR (qRT-PCR)*

qRT-PCR was performed to detect the expression of *PgEXP* genes. Total RNA was isolated from peels by RNA Extraction Kit (FastPure® Plant Total RNA Isolation Kit, Vazyme, Nanjing), and the quality was assessed by electrophoresis and A260/A280. The first-strand cDNA was synthesized from the total RNA by using a cDNA synthesis kit (HiScript III RT SuperMix for qPCR (+gDNA wiper), Vazyme, Nanjing). Specific quantification primers of *PgEXPs* (Table S2) were designed. Pomegranate *PgActin* served as an internal reference gene. The PCR system was 20 μL, including 10 μL Taq Pro Universal SYBR qPCR Master Mix (Vazyme, Nanjing), 0.4 μL upstream and downstream primers, 1 μL cDNA template (the concentration was around 200 ng/μL), and 8.2 μL ddH2O. Three biological replicates for each treatment were to be conducted. The PCR reaction protocol was as follows: 95 °C pre-denaturation for 30 s, 95 °C denaturation for 10 s, 60 °C for 30 s, 40 cycles; the melting curve procedure was as follows: 95 °C 15 s, 60 °C for 30 s, 40 cycles. The relative expression level was analyzed by $2^{-\Delta\Delta\text{CT}}$ method [43]. SPSS 23.0 (CA California, USA) and Origin 2018 software (MA Massachusetts, USA) were used to analyze and plotted the data, respectively.

## 3. Results

### 3.1. Identification and Physicochemical Properties of PgEXP Family Genes

In this study, 33 potential expansin genes were identified from the 'Taishanhong' pomegranate genome. They were named *PgEXP1-PgEXP33* in the order of gene ID to assist later investigation (Table S3). The analysis of physicochemical properties revealed that the coding region of *PgEXPs* ranged from 555 bp (*PgEXP5*) to 1005 bp (*PgEXP31*). The PgEXPs protein contained 185 (*PgEXP5*) to 335 (*PgEXP31*) amino acids, and the protein molecular mass was between 20495.67 kDa (*PgEXP5*) and 36742.26 kDa (*PgEXP31*). The theoretical isoelectric point ranged from 4.55 (*PgEXP15*) to 9.76 (*PgEXP21*) with 81.8% having pI values greater than 7, indicating basic, and the rest less than 7, indicating acidic. The PgEXP's protein instability indices ranged from 18.69 (*PgEXP22*) to 46.86 (*PgEXP7*) with 90.9% having good structural stability. The mean hydrophilic values of PgEXPs protein values ranged from −0.450 (*PgEXP16*) to 0.065 (*PgEXP14*), apart from *PgEXP14*, *PgEXP29*, and *PgEXP30* which were hydrophobic proteins, the others were hydrophilic. EXP usually had a signal peptide sequence at N-terminus. The analysis found that all members except

*PgEXP2*, *PgEXP5*, *PgEXP15*, *PgEXP18*, *PgEXP22*, *PgEXP23,* and *PgEXP33* contained the N-terminal signal peptides, and most signal peptide lengths were around 20 aa. According to the predicted subcellular localization results, all *PgEXPs* were localized in the cell wall.

### 3.2. Phylogenetic Analysis

To investigate the phylogenetic relationships, a phylogenetic tree was constructed using the selected protein sequences of EXPs from pomegranate, *Arabidopsis*, grape, kiwi, and jujube. In the phylogenetic tree (Figure 1), some members of the five species were clustered on one branch, concerning the phylogenetic relationships and naming rules of *AtEXPs*. *PgEXPs* were divided into four subfamilies, namely EXPA, EXPB, EXLA, and EXLB. The size of these four subfamilies varies slightly. The EXPA subfamily was the largest subfamily, with 25 members, while the EXLA subfamily had only one member. The EXPB subfamily had five members and the rest belonged to the EXLB subfamily. The analysis of the phylogenetic tree showed that all five species' EXPs were distributed in four subfamilies, indicating their functions differed. Some genes seemed to diverge early because of their long branches, and they might evolve differently during the long-term phylogeny. Moreover, there were some differences among these five species drawn from the phylogenetic tree, implying that the expansin gene family evolved separately. Through this analysis, we can infer the functions of *PgEXPs* with high similarity from other species' genes with verified functions.

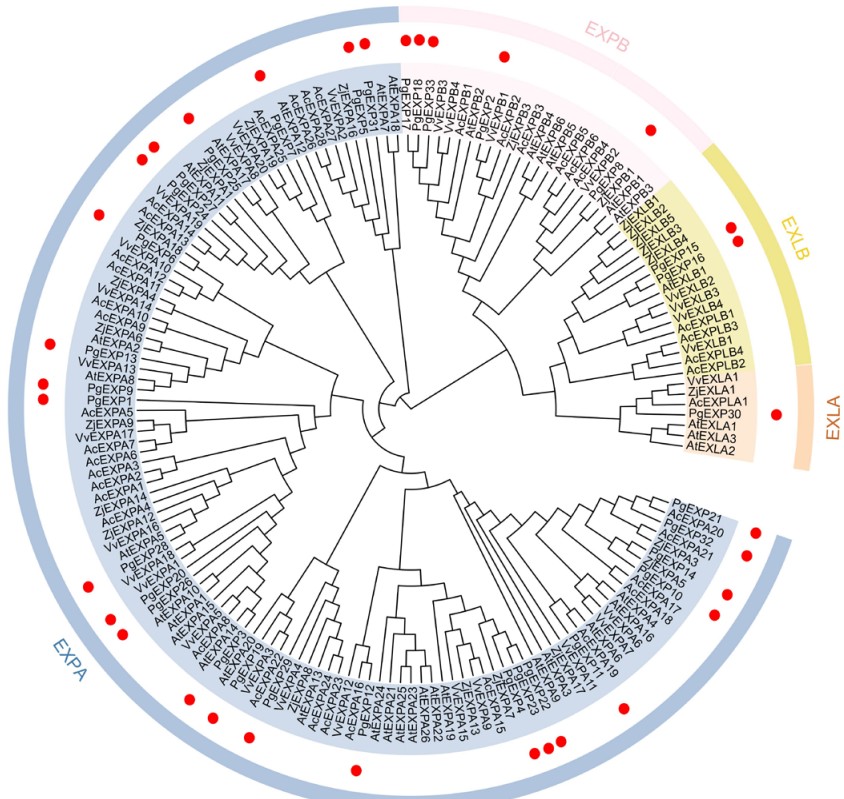

**Figure 1.** Phylogenetic trees in the expansin gene family in pomegranate, *Arabidopsis*, grape, kiwi, and jujube. The phylogenetic tree was constructed using the ML method with 1000 bootstrap replicates. The different colors indicated different subfamilies. The red dots represented *PgEXPs*.

To further compare the quantitative distribution of *EXPs* in different subfamilies, we summarized the number of *EXPs* of 21 species (Table 2), including monocotyledonous plants, dicotyledonous plants, and a nonvascular plant. The results showed that the EXPA subfamily had the largest number of *EXPs* in these species except that in corn (*Zea mays*). The number of EXPA and EXPB subfamilies members was less distinct in

monocotyledonous plants, and the EXLB family members were extremely low. In addition, the number of monocotyledons EXPB subfamily members were significantly higher than that in dicots. The total number of *EXPs* in pomegranate was higher than that in cannabis (*Cannabis sativa*), ginkgo (*Ginkgo biloba*), grape (*Vitis vinifera*), and jujube (*Ziziphus zizyphus*), lower than that in other species within this table.

**Table 2.** Sizes of the four expansin subfamilies in different plants species.

| Species | EXPA | EXPB | EXLA | EXLB | Total | Reference |
|---|---|---|---|---|---|---|
| *Actinidia chinensis* | 28 | 6 | 1 | 4 | 39 | [17] |
| *Arabidopsis thaliana* | 26 | 6 | 3 | 1 | 36 | [7] |
| *Brassica napus* | 79 | 21 | 5 | 4 | 109 | [44] |
| *Brassica rapa* | 39 | 9 | 2 | 3 | 53 | [45] |
| *Cannabis sativa* | 19 | 7 | 1 | 5 | 32 | [18] |
| *Cucumis sativus* | 21 | 3 | 9 | 2 | 35 | [46] |
| *Glycine max* | 49 | 9 | 2 | 15 | 75 | [47] |
| *Ginkgo biloba* | 20 | 1 | 4 | 3 | 28 | [48] |
| *Gossypium hirsutum* | 67 | 12 | 15 | 1 | 93 | [49] |
| *Malus×Domestica* | 34 | 1 | 2 | 4 | 41 | [15] |
| *Nicotiana tabacum* | 36 | 6 | 3 | 7 | 52 | [47] |
| *Oryza sativa* | 34 | 19 | 4 | 1 | 58 | [49] |
| *Physcomitrella patens* | 32 | 0 | 6 | 0 | 38 | [50] |
| *Populus* | 27 | 3 | 2 | 4 | 36 | [19] |
| *Punica granatum* | 25 | 5 | 1 | 2 | 33 | This study |
| *Salix sinopurpurea* | 26 | 3 | 2 | 3 | 34 | [51] |
| *Solanum lycopersicum* | 25 | 8 | 1 | 4 | 38 | [47] |
| *Triticum aestivum* | 26 | 15 | 4 | 0 | 45 | [12] |
| *Vitis vinifera* | 20 | 4 | 1 | 4 | 29 | [14] |
| *Zea mays* | 36 | 48 | 4 | 0 | 88 | [47] |
| *Ziziphus zizyphus* | 19 | 3 | 1 | 7 | 30 | [52] |

*3.3. Analysis of Conserved Domains, Gene Structure, and Protein Conserved Motif*

To further verify the conserved structure of pomegranate *EXPs*, align 33 PgEXP protein sequences and then visualize them with Jalview software. As shown in Figure 2, the protein contained two conserved domains and a signal peptide. The conserved domains were both around 100 aa in length. Of these, domain 1 contained an 'HFD' structure which was part of the glycoside hydrolase-45 (GH45) protein's catalytic site. The 'HFD' structure was conserved in the EXPA and EXPB subfamilies but mutated to 'SFV' and 'DFI' in the EXLA and EXLB subfamilies. Except for *PgEXP28*, other EXPA members characterized the large insertion (α-Insertion) and deletion (α-Deletion). Of the three expansin-like protein sequences, one was classified into the EXLA subfamily based on the presence of a characteristic EXLA extension at the C-terminus with the remaining ones classified into EXLB subfamily. Additionally, *PgEXPs* had a BOX with the sequence signature 'GACGYG' which was missing or mutated in a few members. The variation in conserved sequence features revealed potential functional differences between subfamilies. Other members, with the exception of EXLA, EXLB subfamilies and some EXPA subfamily members (*PgEXP5*, *PgEXP10*, and *PgEXP23*), contained all eight cysteines (Cysteine, C) residues and four conserved tryptophan (tryptophan, W) residues at the hydroxyl terminus. It has been proposed that these sites were located in the expansin's glycosylation and catalytic regions, respectively, that cysteine might play a role in the formation of disulfide bonds, that HFD and conserved aspartate residues might act via a glycosylation mechanism, and that conserved tryptophan might work in the binding of protein and polysaccharide molecules [53].

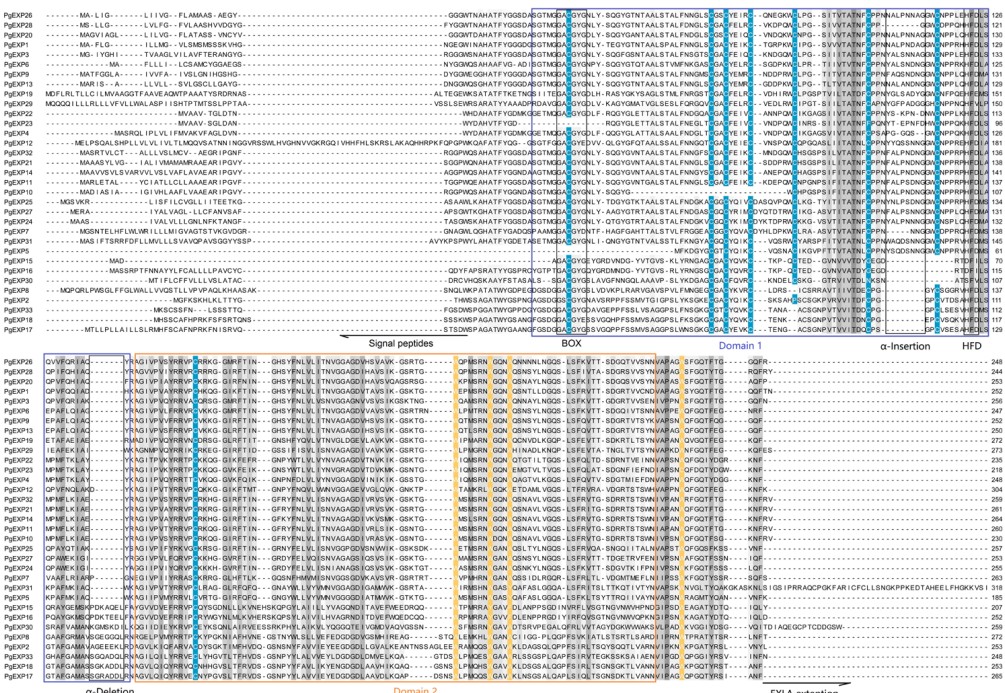

**Figure 2.** Multiple alignments of PgEXPs protein sequencing results. The blue square represented Domain 1. The orange square represented Domain 2. The blue background represented cysteine, and the orange background represented tryptophan.

A total of 10 conserved motifs were identified by the website MEME Suit (Figure 3a). When each motif was submitted to the Pfam server, it was discovered that motif1, motif4, and motif9 encoded conserved domain 1, motif2, motif3, and motif10 encoded the conserved domain 2. Then, the conserved motif distribution of *PgEXPs* was constructed (Figure 3b-B). The results showed that each gene contained 4–8 motifs. Furthermore, all members had motif5 and motif6, localized to essentially the same position, implying that they were the functional basis of pomegranate expansin genes. The others, with unknown functions, were distributed throughout the protein. Meanwhile, because of poor conservatism, motif deletions, additions, or substitutions in individual genes occurred.

To better understand the evolution of the expansin gene family in pomegranate, the exon–intron structures of all identified *PgEXP*s were analyzed. TBtools visualization results (Figure 3b-C) revealed that *PgEXPs* gene structure was relatively simple with 2–5 exons dividing the gene fragment into introns of varying lengths. The EXPA subfamily had three exon–intron structures: seventeen members had three exons and two introns, five members had two exons and one intron, and three members had four exons and three introns. The reason seemed to be the addition or deletion of introns in the structure. The EXPB subfamily had a very stable intron–exon structure as did the EXLB subfamily which had four exons and three introns. Additionally, the EXLA subfamily gene (*PgEXP30*) had the most introns and exons with a gene structure of five exons and four introns.

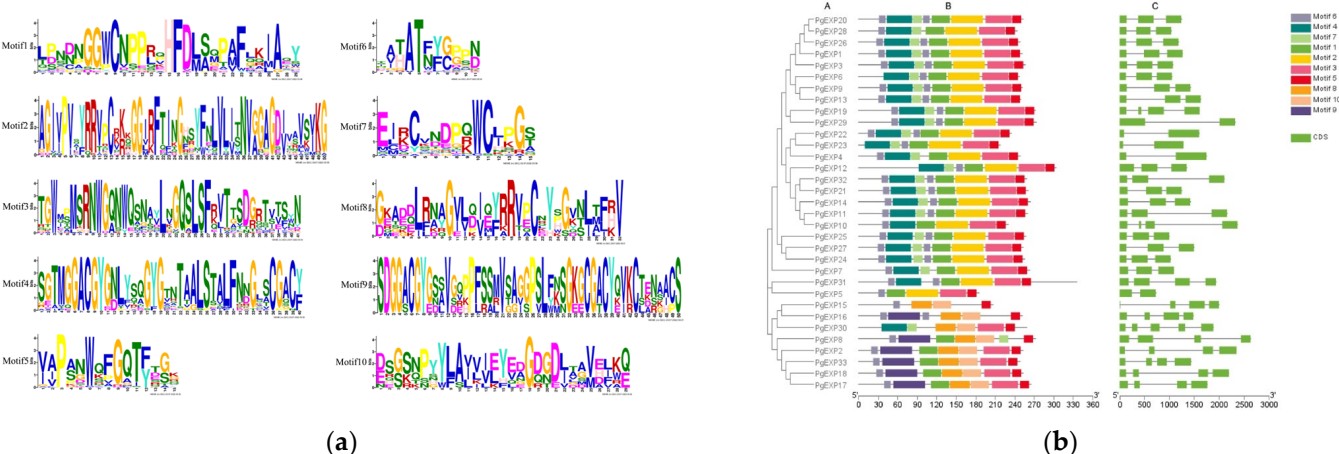

(**a**)                                                 (**b**)

**Figure 3.** (**a**) *PgEXP* genes conserved motifs. The vertical coordinate represented the conserved amino acid and the height of the amino acid letter represented the frequency of occurrence. The horizontal coordinate represented the amino acid's position in the sequence. (**b**) The PgEXP gene family's phylogenetic tree (**A**), conserved motifs (**B**), and gene structure (**C**). Protein motifs in PgEXP members: colored boxes depict the various patterns. The results of phylogenetic analysis were used to perform clustering. Exons and introns were indicated by green boxes and black lines in the gene structure.

### 3.4. Analysis of Cis-Acting Elements

The *cis*-acting elements in the promoter were examined to preferably know the function of *PgEXPs* and the possible regulatory pathways involved. *PgEXPs* contained a total of 36 *cis*-acting elements which were broadly classified into three categories: biotic and abiotic stress responses, plant growth and development, and response elements related to hormone induction (Figure 4). For instance, some *PgEXPs* promoters contained MYB binding sites involved in drought inducibility (MBS), flavonoid biosynthesis genes regulation (MBSI), and light response (MRE). Besides MRE, *PgEXPs* promoter regions contained 11 light responsiveness elements, indicating these *PgEXPs* might be regulated by light. There were also 20 *PgEXPs* that could respond to plant biotic or abiotic stress. Of these, 13 contained the low-temperature responsiveness element LTR, 15 contained the defense and stress responsiveness element TC-rich repeats, and the rest contained the wound-responsive element WUN-motif. In addition, some *PgEXPs* contained anaerobic induction responsiveness elements (ARE) and anoxic specific inducibility responsiveness elements (GC-motif), implying they had a function in the oxygen shunt signaling response. Other regulatory elements were involved in zein metabolism regulation (O2-site), seed-specific regulation (RY-element), and palisade mesophyll cells (HD-Zip 1), suggesting *PgEXPs* related to them might be close to plant growth and development. Furthermore, there were 10 response elements associated with hormone induction. For example, CGTCA-motif and TGACG-motif were both involved in MeJA-responsiveness, but they had different binding sites, CGTCA in the former and TGACG in the latter. All other *PgEXPs*, except *PgEXP3*, *PgEXP29*, and *PgEXP11*, contained at least one phytohormone-responsive element.

### 3.5. Analysis of Protein Interaction Networks

The String protein interaction database was used to predict the co-expression of 33 PgEXP (Figure 5A) proteins (*Arabidopsis thaliana* was chosen as the model species and the AtEXP with the highest similarity was selected). The stronger the contact between the two proteins, the thicker the linkage line. The remaining PgEXPs had no direct contact, indicating that they were not directly controlled. Subsequently, the analysis of the protein interaction network was used to estimate the possible roles of each PgEXP. Among these, PgEXP1, PgEXP3, PgEXP20, and PgEXP26 were found to be identical to EXPA1 (Figure 5B) and were co-expressed with the gibberellin regulatory protein GASA6, the growth regulator At2g22840, and the pectin lyase protein AT5G04310. PgEXP5 and PgEXP31 were identical

to EXPA18 (Figure 5C) and co-expressed with lignin synthesis proteins (RHS19, AT1G30870, and AT3G49960) as well as proteins in the cell wall metabolism-related enzymes production pathway (RHS12, XTH14, and AT5G04960).

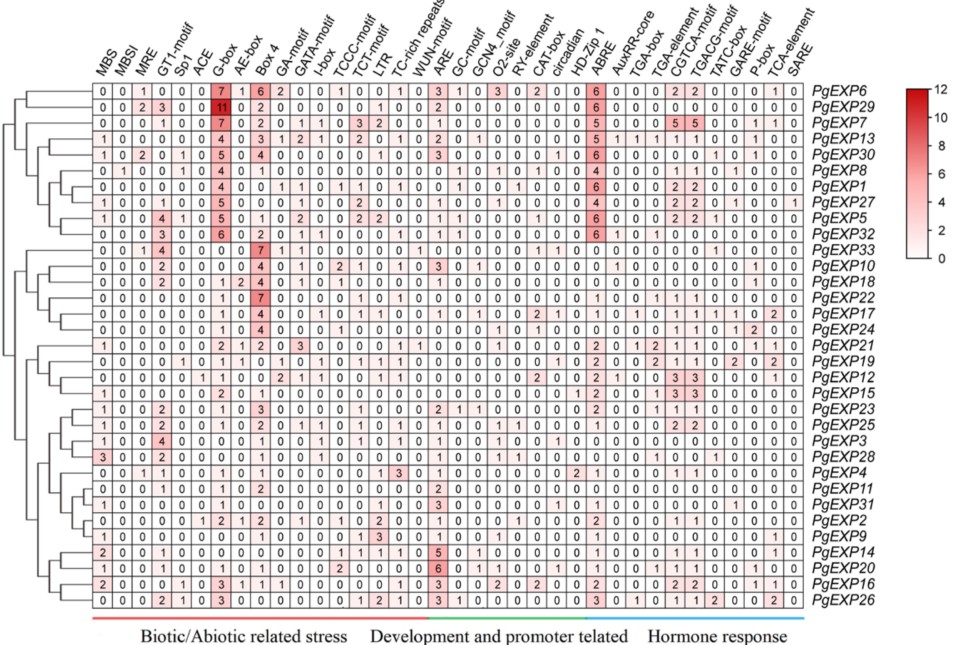

**Figure 4.** Pomegranate *PgEXP*s promoter predicted *cis*-acting elements. The numbers represented the number of *cis*-acting elements. The red line represented the biotic and abiotic related stress in plants. The green line represented plants development and promoter. The blue line represented hormone response related to plants.

### 3.6. Analysis of PgEXPs Gene Expression

To further investigate the gene expression divergence among different tissues, we downloaded RNA-seq data from NCBI for pomegranate (Table 1, Figure 6). Among the 33 *PgEXP* gene expressions in different tissues, ten genes (*PgEXP23, 20, 17, 25, 6, 4, 31, 18, 5,* and *22*) were not expressed or minimally expressed in various tissues. Four genes showed similar expression patterns, *PgEXP33* was expressed at the highest level in the mixed samples of 'Nana', *PgEXP15* was expressed at the highest level in the inner seed coat of 'Tunisia', *PgEXP1* was transcribed at the highest level in the pericarp and outer seed coat of 'Dabenzi', and *PgEXP27* was expressed at the highest level in the inner seed coat of 'Dabenzi'. There were 10 genes (*PgEXP26, 9, 13, 29, 32, 21, 14, 11, 16,* and *30*) that displayed expression in almost all tissues. Of these, *PgEXP26* expression in all tissues was all higher, indicating that it may play a complex function in pomegranate growth and development. Furthermore, we found all genes had low or no expression in two samples which were 5.1–13 mm hermaphrodite and 13.1–25 mm hermaphrodite.

### 3.7. qRT-PCR

To explore the particular roles of *PgEXPs* in fruit development, quantitative real-time PCR (qRT-PCR) was used to examine the expression patterns of *PgEXPs* in pomegranate pericarp. Finally, we chose 24 *PgEXP* and assessed their expression at 6 periods of 'Daqing-pitian' pomegranate pericarp (Figure 7). In general, *PgEXP1, PgEXP26, PgEXP28,* and *PgEXP32* expression levels decreased from P1 to P6; of these, *PgEXP28* was low or almost not expressed in P2-P6, and expression levels were considerably greater in P1 than in P2–P6. *PgEXP23* expression, on the other hand, rose gradually across all periods, peaking in P6. *PgEXP23* was found to be more abundant in the later phases of fruit growth as well, suggesting that it was important for fruit ripening. *PgEXP3* and *PgEXP13* had the highest expression levels at P5, *PgEXP15* had the highest expression levels at P2, *PgEXP9*

had higher expression levels at both P2 and P5, and the expression levels of *PgEXPs* varied considerably at different phases of fruit development.

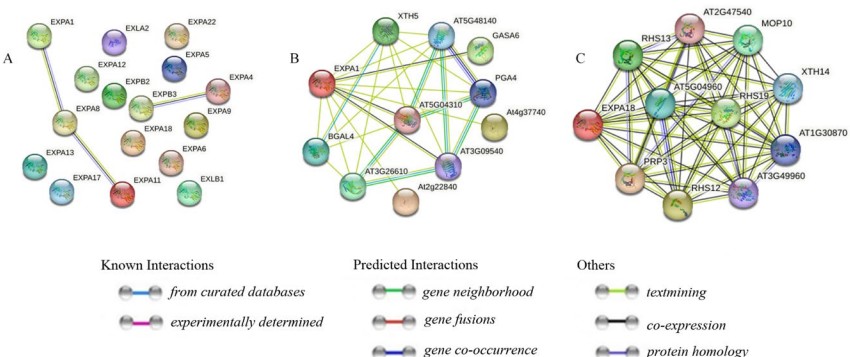

**Figure 5.** PgEXP proteins (**A**), PgEXP1, PgEXP3, PgEXP20, and PgEXP26 protein (**B**), PgEXP5 and PgEXP31 protein (**C**) functional interaction network and gene co-expression diagram. XTH5: Probable xyloglucan endotransglucosylase/hydrolase protein 5; BGAL4: Beta-galactosidase 4; AT5G48140: Galacturan 1,4-alpha-galacturonidase; AT5G04310: Pectin lyase-like superfamily protein; AT3G26610: Pectin lyase-like superfamily protein; GASA6: Gibberellin-regulated family protein; PGA4: Galacturan 1,4-alpha-galacturonidase; At4g37740: Growth-regulating factor 2; AT3G09540: Pectin lyase-like superfamily protein; At2g22840: Growth-regulating factor 1; RHS13: Root hair specific 13; PRP3: Arabidopsis thaliana proline-rich protein 3; AT5G04960: Plant invertase/pectin methylesterase inhibitor superfamily; MOP10: Pollen Ole e 1 allergen and extensin family protein; AT1G30870: Peroxidase superfamily protein; AT3G49960: Peroxidase superfamily protein.

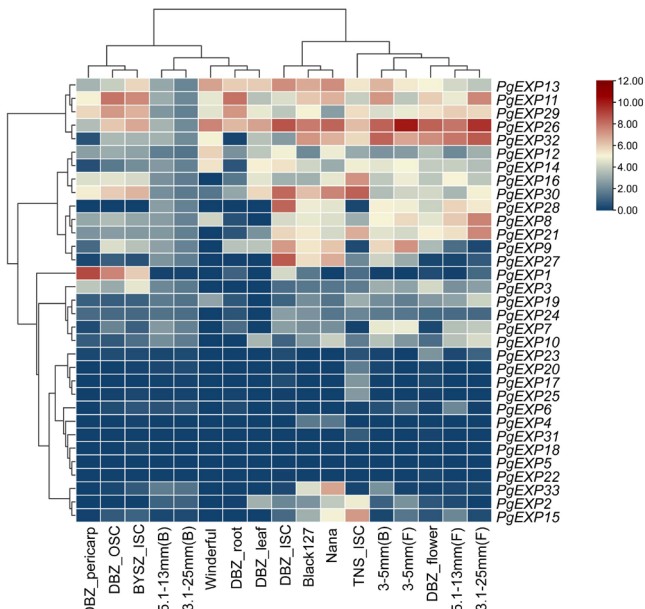

**Figure 6.** Heat map of *PgEXP* gene expression in pomegranate tissues. DBZ_pericarp: 'Dabenzi' Pericarp; DBZ_OSC: 'Dabenzi' Outer seed coat; BYSZ_ISC: 'Baiyushizi' Inner seed coat; 5.1–13 mm(B): 'Tunisia' Bisexual flowers (5.1–13.0 mm); 13.1–25 mm(B): 'Tunisia' Bisexual flowers (13.1–25.0 mm); Wonderful: 'Wonderful' Pericarp; DBZ_root: 'Dabenzi' Root; DBZ_leaf: 'Dabenzi' Leaf; DBZ_ISC: 'Dabenzi' Inner seed coat; Black127: 'Black127' Mix of leaves, flowers, fruit and roots; Nana: 'Nana' Mix of leaves, flowers, fruit and roots; TNS_ISC: 'Tunisia' Inner seed coat; 3–5 mm(B): 'Tunisia' Bisexual flowers (3–5 mm); 3–5 mm(F): 'Tunisia' Functional male flowers (3–5 mm); DBZ_flower: 'Dabenzi' Flower; 5.1–13 mm(F): 'Tunisia' Functional male flowers (5.1–13.0 mm); 13.1–25 mm(F): 'Tunisia' Functional male flowers (13.1–25.0 mm).

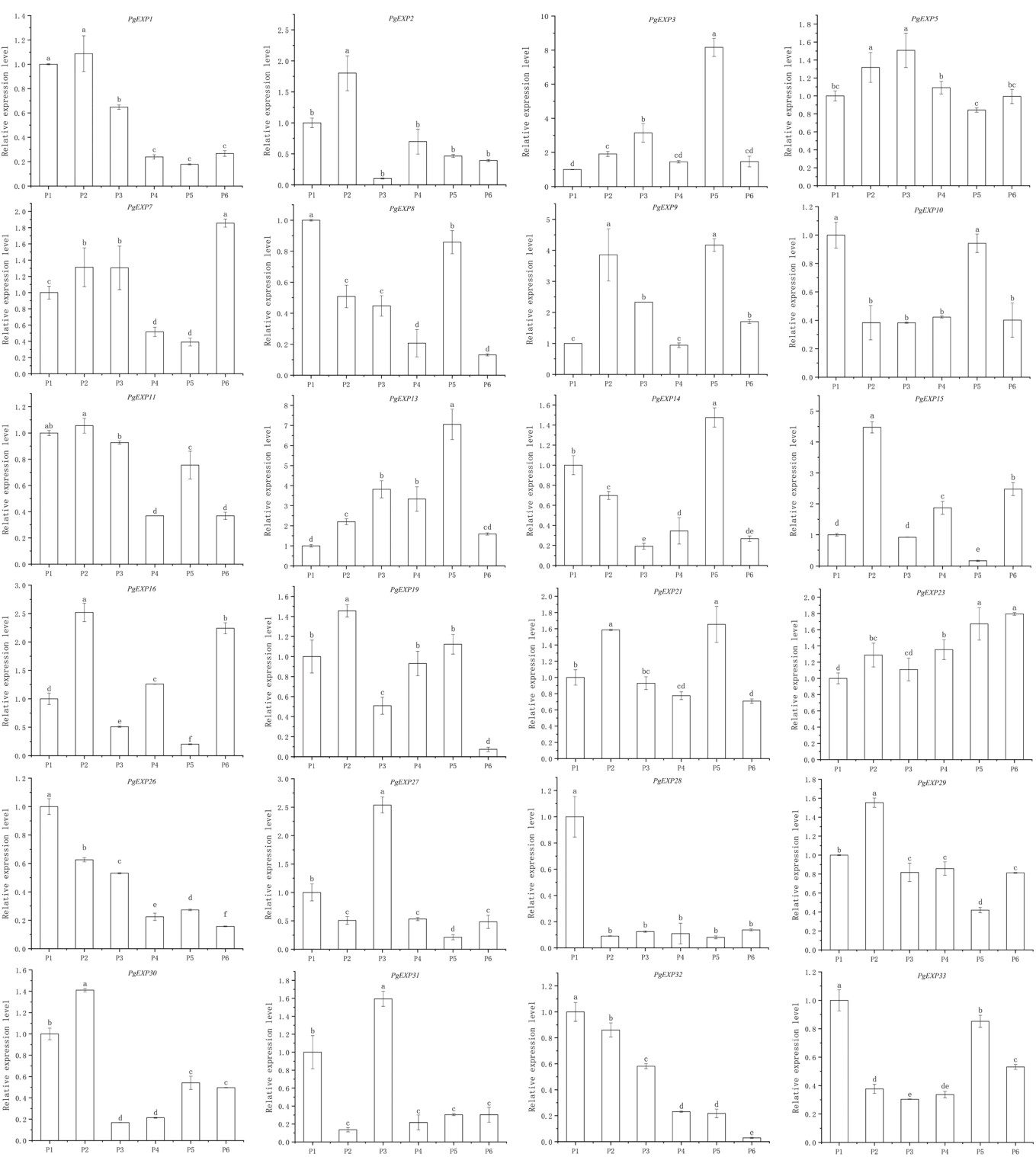

**Figure 7.** The expression patterns of 24 *PgEXPs* in pomegranate peel at 6 developmental periods obtained by qRT-PCR analysis. The dates were 25 August, 3 September, 12 September, 21 September, 30 September, and 9 October designated as P1~P6, respectively. The data shown represent an average of three independent experiments ± SD. The vertical bars show the standard error. Bars with different letters (a–f) indicate significant differences at *p* < 0.05 according to Duncan's test.

## 4. Discussion

Since cell wall enlargement played a key role in plant morphogenesis, the mechanism of cell wall extension had been a focus of investigation [2]. During turgor-mediated growth, cell wall stress relaxation occurred [54] which involved cell wall-loosening factors, such as EXPs [55]. Expansins might offer a mechanism for cell wall elongation by interfering with the binding of microfibrils to the cell wall [4]. The expansin gene family has been thoroughly discovered in numerous plants [17,18]. Previously, systematic analyses were carried out to identify and characterize expansin families in a variety of models, crop and fruit plant species, such as tomatoes [56], pepper [57], litchi [58], and apple [59]. Studies, subsequently, showed that a close relationship was established between expansins and fruit cracking. Fruit cracking tremendously damages the appearance of the fruit, easily leads to pathogen invasion, greatly reduces marketability, and causes immense economic losses. Currently, pomegranate fresh fruit, seedlings, and processing industry have a great promising future, while fruit cracking is a major cause of fruit loss in pomegranate [60]. Thus, it is considerable and urgent to explore the roles expansins play in pomegranates.

In this work, we first identified 33 pomegranate expansins through genome-wide analysis and then used web-based tools and resources to predict the properties of these proteins, such as isoelectric point and molecular weight. The prediction of subcellular localization showed that all genes localized in the cell wall. These helped us learn more about these genes.

Based on the genes of each ancestor, plant expansins experienced varying degrees of gene duplication and expansion [6]. The dicotyledonous plant expansin gene family, which included *Arabidopsis*, soybean [61], and watermelon [62], was mostly extended via tandem and segmental replication. According to the established EXPs classification of pomegranate, *Arabidopsis*, grape, kiwi, and jujube, *PgEXPs* were divided into four subfamilies, EXPA, EXPB, EXLA, and EXLB. In addition, we found that the large branches of the phylogenetic tree usually contained *EXPs* of different species, while on some small branches, there were usually only *EXPs* of the same species, suggesting that EXPs amplification had occurred before the differentiation of these species and after the differentiation of the species, EXPs amplification occurred again. This view was corroborated by the phylogenetic analysis of tobacco [47], soybean [47], and rice [49]. Analysis and comparison of the sizes of expansin subfamilies in 21 species revealed an uneven distribution of each gene subfamily among species. For example, the EXPB subfamily members were significantly more numerous in momocots than that in dicots, with 19 in rice, 15 in wheat, and 48 in maize, while around 10 in dicots (Table 2), it may be due to the greater expansion and retention of gene duplication events in monocots [61].

The architecture of various *PgEXP* genes varied with minimal variation in the number of exons and a substantial diversity in the length of introns. The analysis of gene structure revealed that *PgEXPs* had 2–5 exons which was compatible with the number of exons found in grapes [15] and land cotton [63], showing that the pomegranate expansin gene structure was largely conserved. According to the conserved motif analysis, all *PgEXPs* contained a total of 10 motifs with a similar distribution of motifs within the same subgroup. The high degree of sequence identity and similar gene structure of *PgEXPs* within each family indicated that the pomegranate expansin family had undergone gene duplication throughout evolution, resulting in multiple copies with a partial or complete overlap in function.

*Cis*-acting elements offer genes the ability to work in developmental or environmental regulation. Light responsiveness elements were the first class of enriched *cis*-acting elements in *PgEXPs*. Of these, G-box was the most abundant light responsiveness element. In *PgEXPs* promoter regions, several elements associated with plant growth, environmental stress, and hormone induction existed. The results were consistent with the findings of that in cannabis [18] and *Panax ginseng* [64], indicating the regulatory elements of the *EXPs* were more conserved across species. For instance, the promoter regions contained many *cis*-acting elements engaged in seed-specific regulation, zein metabolism regula-

tion, and palisade mesophyll cell differentiation, suggesting that these *PgEXPs* worked in pomegranate seed germination and leaf development [65]. Additionally, MYB binding sites were found in the promoter regions of 20 *PgEXPs*. A total of fifteen *PgEXPs* had binding sites (MBS) implicated in drought-inducibility which might be related to plants' biotic and abiotic stress. Five *PgEXPs* had sites (MRE) involved in light responsiveness and there was also one site (MBSI) implicated in the regulation of flavonoid biosynthesis genes. These MYB protein binding sites could be detected in the sour cherry expansin gene family as well [66]. Furthermore, the findings of hormone-inducing *cis*-acting elements, such as abscisic acid, MeJA, gibberellin, and salicylic acid, implied that *PgEXPs* might be triggered by a variety of hormone-signaling molecules [67].

The protein interaction network of PgEXPs revealed that some genes had cocations. The prediction results revealed that PgEXP1, PgEXP3, PgEXP20, and PgEXP26 were co-expressed with pectin lyase proteins. Combined with the expression level of these genes, we found that PgEXP26 might be involved in pectin lyase protein production of flower, leaf, pericarp, and mixed tissues. In addition, PgEXP1 expression levels in the inner seed coat of 'Dabenzi' and 'Baiyushizi' were greater than that in 'Tunisia', it might be owing to genotype differences. According to further analysis of the expression pattern of *PgEXPs* in different tissues, *PgEXP30*, a member of the EXLA subfamily, was expressed in a higher level in flower, root, and mixed sample than other tissues. In contrast, two members of the EXLB subfamily, *PgEXP15* and *PgEXP16*, were exclusively expressed in 'Dabenzi' root tissues, implying that EXLB played a role in pomegranate root growth and development. Furthermore, some genes were not or weakly expressed in all tissues, indicating that they were not engaged in pomegranate growth or stress regulation. Finally, qRT-PCR was used to examine the expression pattern of *PgEXPs* in pomegranate pericarp. The expression level of *PgEXP23* rose consistently with time advancement, indicating that *PgEXP23* may play an important role in fruit ripening. *PgEXPs* expression levels varied considerably across fruit developmental phases, suggesting that *PgEXPs* genes may have diverse functions.

## 5. Conclusions

This work was the first comprehensive genome-wide analysis of the pomegranate expansin gene family. We identified 33 *PgEXPs* from the pomegranate 'Taishanhong' genome. Based on a phylogenetic tree of pomegranate, *Arabidopsis*, grape, kiwi, and jujube expansin genes, pomegranate genes were split into four subfamilies, EXPA (25 members), EXPB (5 members), EXLA (1 member), and EXLB (2 members). Members of subfamilies were highly conserved in motif and gene structure. Analysis of tissue-specific expression patterns of *PgEXPs* revealed that they may function differently in regulating organ/tissue morphology formation and development. Analysis of promoter cis-acting elements revealed that *PgEXPs* may respond to development and stress. qRT-PCR results played a role in exploring the function of *PgEXP* genes in pomegranate. The above analysis added to our knowledge of the expansin gene family in pomegranates and provided insights into the possible functional involvement of pomegranate expansin genes.

**Supplementary Materials:** The following supporting information can be downloaded at: https://www.mdpi.com/article/10.3390/horticulturae9050539/s1, Table S1: Referenced EXPs used to contruct phylogenetic tree, Table S2: Primers for qRT-PCR, Table S3: Basic information of pomegranate gene family, Table S4: The cis-acting elements of PgEXPs.

**Author Contributions:** Conceptualization, X.X. and Z.Y.; methodology, X.X. and Z.Y.; software, X.X. and Y.W.; visualization, X.X. formal analysis, X.X.; writing—original draft preparation, X.X.; writing—review and editing, Y.W., X.Z. and Z.Y.; supervision, Z.Y.; funding acquisition, X.Z. and Z.Y. All authors have read and agreed to the published version of the manuscript.

**Funding:** This work was supported by National Natural Science Foundation of China (31901341) and the Priority Academic Program Development of Jiangsu High Education Institutions [PAPD].

**Data Availability Statement:** The data presented in this work are available upon request from the corresponding author and the public pomegranate transcriptomes presented in this work are available in the insert article.

**Conflicts of Interest:** The authors declare no conflict of interest.

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
