# Peer review of "Uncovering the Expansin Gene Family in Pomegranate (Punica granatum L.): Genomic Identification and Expression Analysis"

_horticulturae, doi:10.3390/horticulturae9050539_

Round 1

Reviewer 1 Report

The authors have been written this MS following the findings.  I am quite satisfied to see the experimental design, results and their explanations. Rephrase the Introduction and Discussion parts to improve the English, in my perspective. However, I strongly recommend to work on the following concern as-

Line 30-36, 63-65 and 68-85: Rewite the sentences to meet scientific standard.

Line 135: the section 2.5: RNA-seq analysis is very short that needs to clearify a bit more details

In Table 3; protein localization is in the cell wall for all cases. You should remove from the table and write in a sentence in the 3.1 paragraph

Figure 1, 2, and 6 are very difficult to read specially. Need to make well visible. Wish you the best.

Author Response

Response to Reviewer 1 Comments

Point 1: Line 30-36, 63-65 and 68-85: Rewite the sentences to meet scientific standard.

Response 1: I have rewited this sentences.

Line 30-36: replace “The cell wall is an essential and distinct structure in plant cells. It determines cell shape and size, provides mechanical support and stiffness, and serves as the cell's last barrier against pathogens as well. Owing to the importance of cell wall enlargement for plant morphogenesis, it is becoming a hot focus to study the mechanism of cell wall extension. Previously, a process called the 'acid growth phenomenon', has shown that in an acidic environment, cell wall can be extended without structural changes, and the extending characteristics can be induced or inhibited in a short time.” with “In plant cells, the cell wall is an essential and distinct structure. It determines cell sharp and size, provides mechanical support and stiffness, and is the cell’s first barrier against pathogens. Because of the importance of cell wall enlargement in plant morphogenesis, it is becoming a hot focus to study the mechanism of cell wall extension. Previously, the 'acid growth phenomenon' shown that in an acidic environment, the cell wall can be extended without structural changes, and the extending characteristics can be induced or inhibited in a short time.”

Line 63-65: replace ” Overexpression of the FaEXP2 gene increased pectin content in transgenic lines' cell walls and decreased expression levels of genes encoding cell wall degrading enzymes, causing the fruit to be hard or late ripening. The expression of the expansin gene was significantly decreased in crack-prone longan pericarp, showing the gene plays a crucial role in crack resistance creation.” with “Overexpressing FaEXP2 gene increased pectin content in the cell walls of transgenic lines while decreasing expression levels of genes encoding cell wall degrading enzymes, resulting in hard or late ripening fruit. The expression of the expansin gene was significantly decreased in crack-prone longan pericarp, showing that the gene played a crucial role in crack resistance creation.”

Line 68-85: replace “Pomegranate (Punica granatum L.) is an economically important tree species and widely planted worldwide, native to Central Asia, including Iran, Afghanistan,and the Caucasus. It is well known for its vivid red skin and juicy seeds. Furthermore, the fruit peel and juice extracts are rich in antioxidants, such as polyphenols, and have been suggested to have advantageous effects in cardiovascular, tumors, diabetes, and other diseases. In recent years, scholars have successively assembled several pomegranate genomes, which are 'Taishanhong', 'Dabenzi',and 'Tunisia', and acquired high-quality genome maps, offering an essential molecular biological basis for pomegranate genetic improvement. With advances in molecular biology, not only can we study specific gene family functions bioinformatically, but also analyze genes how to regulate plant growth and development, environmental stress. What’s more, these progresses play important roles in revealing the mechanism of development and stress tolerance. The research to date on pomegranate expansins (PgEXP) is still in its early stage. Thus, based on the 'Taishanhong' genome, we combined bioinformatic approaches to identify members of the pomegranate expansin gene family, and analyzed their physicochemical properties, conserved domain, evolutionary relationship, cis-acting elements and tissue organ expression. Our study will lay a foundation for further research on the functions of the expansin gene in pomegranate.” with “Pomegranate (Punica granatum L.) is a species of economically important tree that is widely planted worldwide, native to Central Asia, including Iran, Afghanistan, and the Caucasus. It is well known for its vivid red skin and juicy seeds. Furthermore, the fruit peel and juice extracts are rich in antioxidants like polyphenols, which have been suggested to have positive effects on diabetes, cardiovascular, tumors, and other diseases. In recent years, scholars have successively assembled several pomegranate genomes, including 'Taishanhong', 'Dabenzi', and 'Tunisia', and acquired high-quality genome maps, offering an essential molecular biological basis for pomegranate genetic improvement. With advancements in molecular biology, we may not only study specific gene family functions bioinformatically, but also analyze genes that regulate plant growth and development, as well as environmental stress. What’s more, these progresses play important roles in revealing the mechanism of development and stress tolerance. The research to date on pomegranate expansins (PgEXP) is still in its early stage. Thus, based on the 'Taishanhong' genome, we used bioinformatic approaches to identify members of the pomegranate expansin gene family, and analyzed their physicochemical properties, conserved domain, evolutionary relationship, cis-acting elements and tissue organ expression. Our study will lay a foundation for further research on the functions of the expansin gene in pomegranate.”

Point 2: Line 135: the section 2.5: RNA-seq analysis is very short that needs to clearify a bit more details

Response 2: I have added more details in this section. The details are as following: To analyze expression patterns of PgEXPs in different pomegranate tissues and organs, the published transcriptome data of six pomegranate varieties (‘Dabenzi’, ‘Tunisia’, ‘Baiyushizi’, ‘Black127’, ‘Nana’, and ‘Wonderful’) were downloaded from NCBI (http://www.ncbi.nlm.nih.gov/, access on 10 February 2023), including outer seed coat, inner seed coat, pericarp, flower, root, leaf, and mixed samples of roots, leaf, flower, and fruit, as shown in Table 1. Then the transcriptomic data were calculated and analyzed with Kallisto v0.44.0 software (California, USA), and the resulting values were transformed into Log2(TPM+1) (Table S1) and finally the expression heatmap was created by TBtools.

Point 3: In Table 3; protein localization is in the cell wall for all cases. You should remove from the table and write in a sentence in the 3.1 paragraph

Response 3: I have added it in paragraph and move the Table S3 into supplementary materias.

Point 4: Figure 1, 2, and 6 are very difficult to read specially. Need to make well visible. Wish you the best.

Response 4: I have replaced all figures with higher resolution versions, and put them in supplementary materias for viewing.

Reviewer 2 Report

1. General:

- The Pomegranate scientific name (Punica granatum L.) was indicated just once (line 68). It is recommended to add it in the title and/or materials and methods because scientific name is the bases of any study to identify the organism under investigation.

- The manuscript language is fair but needs many corrections, please check the entire manuscript for weak phrases and common structural mistakes. Few correction were mentioned for the "Abstract", please see below.

2. Abstract:

- Line 11: replace "in environmental resistance" with "under environmental stresses".

- Line 12: replace "lots" with "many".

- Line 12: italicize "EXPs" as it refers to genes in this case.

- Lines 12-13: "but none in pomegranate" is incorrect statement for expansin genes, see for example:

Jeong, H. J., Park, M. Y., & Kim, S. (2018). Identification of chromosomal translocation causing inactivation of the gene encoding anthocyanidin synthase in white pomegranate (Punica granatum L.) and development of a molecular marker for genotypic selection of fruit colors. Horticulture, Environment, and Biotechnology59, 857-864.

- Line 17: replace "elements" with "element".

- Line 17: replace "plenty" with "many".

- Lines 21-22: replace "Real-time fluorescence quantification" with "quantitative real-time PCR".

- Line 24: "fruit ripening"

- Line 24: replace "Contrary to this" with " On the other hand".

3. Introduction:

- Line 47: replace "What's more" with "Moreover".

- Lines 47-48: replace "acted upon" with "act upon", it is a fact now.

- Line 49: replace " they do worked" with "they play a major role".

- Line 53: replace "the sequence" with "their sequence", AND Line 54: remove "of them".

- Line 55: replace "regulated " with "regulate", these are facts now.

- Line 57: replace "ZmEXPB13 was an" with " ZmEXPB13, an".

- Sill too many linguistic mistakes!!! …..Please revise!

3. Materials and Methods:

- Lines 95-110: you need to indicate access data for each site.

5. Materials and Methods:

- Lines 88-89: please make sure you are not utilizing unpublished third party RNA-seq data, otherwise you need a permission. In addition, the authors should be cautious when interpreting these results. In fact, the data cannot be compared with each other as RNA-seq from multiple genotypes were utilized to present the expression levels, which for sure can be far away from reality!!!

- Line 114: Neighbour Joining is not a phylogenetic method, but rather a phenetic one. It establishes relationships between sequences according to their genetic distance without taking into account an evolutionary model. On the contrary Maximum Likelihood method uses more complex evolution model, which is stronger than Neighbour Joining. In addition, Maximum Likelihood or Maximum Parsimony, can figure out the amount of change between sequences and their nature. Therefore, it is always recommend to use the most robust phylogeny based on Maximum Likelihood or Maximum Parsimony rather than Neighbour Joining for publications. Please see Molecular Phylogenetics and Evolution, 2(1), 1-5.

- Line 137: you should type TPM in full the first time it is mentioned in text.

- Lines 143-146: Utilized kits and protocols for RNA isolation and cDNA showed be mentioned.

- Lines 146-148: it is of no use to mention PCR component volumes but rather their concentrations!

- Line 153: it is recommended to transfer table (2) with PCR primers to supplementary data. It is of limited use and is taking a big part of the manuscript.

6. Results:

- Line 173: it is recommended to transfer AND join table (3) into table (S1) in supplementary data. In fact, several entries, e.i. Gene Name, Gene ID, MW, pi, Instability index and GRAVY are duplicated in both tables (3) and (S1)!!! And again this will save huge section from the manuscript.

- Line 191: Please replace figure (1) with a higher resolution version.

- Line 213: Please replace figure (2) with a higher resolution version.

- Line 227: Please replace figure (3) with a higher resolution version.

- Line 267: Please replace figure (4) with a higher resolution version.

- Line 285: Please replace figure (5) with a higher resolution version.

- Lines 296-331: Again, the authors should be cautious when interpreting these results. In fact, the data cannot be compared with each other as RNA-seq from multiple genotypes were utilized to present the expression levels, which for sure can be far away from reality!!!

- Line 337: Please replace figure (7) with a higher resolution version.

- Lines 140-142: you mentioned that you used "SE" for error bars in qRT-PCR data. This is not correct as the data resembles  relative values as compared to the control (in your case P1). However, you can employ the SE to calculate the 95% confidence interval (95% CI, z score = 1.96), where C.I. = Mean ΔΔCt ± (z x SE). Please consult the following articles:

Wang, B., Tian, S., Zhou, Q., & Zeng, X. (2006). Did Your RNAi Experiment Work. Reliably Validating RNA Interference with qRT-PCR.

Sadder, M. T., & Al-Doss, A. A. (2014). Characterization of dehydrin AhDHN from Mediterranean saltbush (Atriplex halimus). Turkish Journal of Biology, 38(4), 469-477.

7. Discussion:

- A close relationship was established between expansins and fruit cracking in important crops including tomatoes (Jiang et al 2019), pepper (Liu et al 2022) and several other fruit crops (Wang et al., 2021). Likewise, fruit cracking is a major cause of fruit loss in pomegranate (Singh et al., 2020; Wang et al., 2022). However, nothing was mentioned about that in the manuscript, therefore, please add a paragraph about pomegranate cracking and expansins and discuss that.

Jiang, F., Lopez, A., Jeon, S., de Freitas, S. T., Yu, Q., Wu, Z., ... & Mitcham, E. (2019). Disassembly of the fruit cell wall by the ripening-associated polygalacturonase and expansin influences tomato cracking. Horticulture research6.

Liu, Y. L., Chen, S. Y., Liu, G. T., Jia, X. Y., ul Haq, S., Deng, Z. J., ... & Gong, Z. H. (2022). Morphological, physiochemical, and transcriptome analysis and CaEXP4 identification during pepper (Capsicum annuum L.) fruit cracking. Scientia Horticulturae297, 110982.

Wang, Y., Guo, L., Zhao, X., Zhao, Y., Hao, Z., Luo, H., & Yuan, Z. (2021). Advances in mechanisms and omics pertaining to fruit cracking in horticultural plants. Agronomy11(6), 1045.

Singh, A., Shukla, A. K., & Meghwal, P. R. (2020). Fruit cracking in pomegranate: extent, cause, and management–A Review. International Journal of Fruit Science20(sup3), S1234-S1253.

Wang, Y., Zhao, Y., Wu, Y., Zhao, X., Hao, Z., Luo, H., & Yuan, Z. (2022). Transcriptional profiling of long non-coding RNAs regulating fruit cracking in Punica granatum L. under bagging. Frontiers in Plant Science13.

8.Conclusions:

Lines 408-417: these are repeated results not conclusion, please remove and add your own conclusions, you presented tremendous data!

9. References:

- It is recommended to incorporate more RECENT articles as only one fifth (11 out from 51) of  cited articles were published in the last five years.

Author Response

Response to Reviewer 2 Comments

Point 1: The Pomegranate scientific name (Punica granatum L.) was indicated just once (line 68). It is recommended to add it in the title and/or materials and methods because scientific name is the bases of any study to identify the organism under investigation.

Response 1: I have added it in the title. 

Uncovering the Expansin Gene Family in Pomegranate (Punica granatum L.): Genomic Identification and Expression Analysis

Point 2: Sill too many linguistic mistakes!!! …..Please revise!

Response 2: Thank you for the correction and I have tried my best to revise it.

Point 3: Lines 95-110: you need to indicate access data for each site.

Response 3: I have added it for each site.

Point 4: Lines 88-89: please make sure you are not utilizing unpublished third party RNA-seq data, otherwise you need a permission. In addition, the authors should be cautious when interpreting these results. In fact, the data cannot be compared with each other as RNA-seq from multiple genotypes were utilized to present the expression levels, which for sure can be far away from reality!!!

Response 4: This question may be raised because I have misrepresented. I ensure that these RNA-seq data are published. In this paragraph, I would like to know the expression levels of pomegranate expansin genes in different tissues, and whether they were possible to play a role in plant growth and development.

The following articles were consulted: Zhang X, Wang S, Ren Y, Gan C, Li B, Fan Y, Zhao X, Yuan Z. Identification, Analysis and Gene Cloning of the SWEET Gene Family Provide Insights into Sugar Transport in Pomegranate (Punica granatum). International Journal of Molecular Sciences. 2022; 23(5):2471.

Dal-santo, S.; Vannozzi, A.; Tornielli, G.B.; Fasoli, M.; Venturini, L.; Pezzotti, M.; Zenoni, S. Genome-wide analysis of the expansin gene superfamily reveals grapevine-specific structural and functional characteristics. PLoS One. 2013, 8 (4): e62206.

Point 5: Line 114: Neighbour Joining is not a phylogenetic method, but rather a phenetic one. It establishes relationships between sequences according to their genetic distance without taking into account an evolutionary model. On the contrary Maximum Likelihood method uses more complex evolution model, which is stronger than Neighbour Joining. In addition, Maximum Likelihood or Maximum Parsimony, can figure out the amount of change between sequences and their nature. Therefore, it is always recommend to use the most robust phylogeny based on Maximum Likelihood or Maximum Parsimony rather than Neighbour Joining for publications. Please see Molecular Phylogenetics and Evolution, 2(1), 1-5.

Response 5: I have revised it to build a maximum likelihood tree with IQTree2 V2.1.3 under the best-fitting model with 1000 bootstrap replicates. The selected amino acid sequences of EXPs from pomegranate, Arabidopsis thalianaVitis vinifera, Actinidia chinensis, Ziziphus ziziphus.

Point 6: You should type TPM in full the first time it is mentioned in text.

Response 6: I have added it and the Table S1 was put in supplementary materias.

Point 7: Utilized kits and protocols for RNA isolation and cDNA showed be mentioned.

Response 7: I have added them in the article. Total RNA was isolated from fruit peels by RNA Extraction Kit (FastPure® Plant Total RNA Isolation Kit, Vazyme, Nanjing), and the quality was assessed by electrophoresis and A260/A280. The first-strand cDNA was synthesized from the total RNA by using cDNA synthesis kit (HiScript III RT SuperMix for qPCR (+gDNA wiper), Vazyme, Nanjing).

Point 8: It is of no use to mention PCR component volumes but rather their concentrations!

Response 8: I have added them in the article.The PCR reaction system was 20 μL, including 10 μL Taq Pro Universal SYBR qPCR Master Mix (Vazyme, Nanjing), 0.4 μL upstream and downstream primers, 1 μL cDNA template (the concentration was around 200 ng/μL) and 8.2 μL ddH2O.

Point 9:  It is recommended to transfer table (2) with PCR primers to supplementary data. It is of limited use and is taking a big part of the manuscript.

Response 9: I have moved it into supplementary materias.

Point 10: It is recommended to transfer AND join table (3) into table (S1) in supplementary data. In fact, several entries, e.i. Gene Name, Gene ID, MW, pi, Instability index and GRAVY are duplicated in both tables (3) and (S1)!!! And again this will save huge section from the manuscript.

Response 10: I have moved it into supplementary materias.

Point 11: Please replace figure with a higher resolution version.

Response 11: I have replaced them with higher resolution versions and put in supplementary materias for viewing.

Point 12: You mentioned that you used "SE" for error bars in qRT-PCR data. This is not correct as the data resembles  relative values as compared to the control (in your case P1). However, you can employ the SE to calculate the 95% confidence interval (95% CI, z score = 1.96), where C.I. = Mean ΔΔCt ± (z x SE).

Response 12: First of all, thank you for pointing out this problem. When I analyzed this section, I used the experimental methods from the following articles, which was similar with my experiment and have been published in recent years. The expression patterns of genes obtained by the real-time quantitative PCR analysis. Data shown represent an average of three independent experiments ± SD. The vertical bars show the standard error. I will try this method you propose in the next experimentst.

The following articles were consulted:

Wang Y, Guo L, Zhao Y, Zhao X, Yuan Z. Systematic Analysis and Expression Profiles of the 4-Coumarate: CoA Ligase (4CL) Gene Family in Pomegranate (Punica granatum L.). International Journal of Molecular Sciences. 2022; 23(7):3509.

Zhang X, Wang S, Ren Y, Gan C, Li B, Fan Y, Zhao X, Yuan Z. Identification, Analysis and Gene Cloning of the SWEET Gene Family Provide Insights into Sugar Transport in Pomegranate (Punica granatum). International Journal of Molecular Sciences. 2022; 23(5):2471.

Hou, L., Zhang, Z., Dou, S. et al. Genome-wide identification, characterization, and expression analysis of the expansin gene family in Chinese jujube (Ziziphus jujuba Mill.). Planta 249, 815–829 (2019).

Zhang, S., Xu, R., Gao, Z. et al. A genome-wide analysis of the expansin genes in Malus × Domestica . Mol Genet Genomics. 289, 225–236 (2014).

Point 13:  A close relationship was established between expansins and fruit cracking in important crops including tomatoes (Jiang et al 2019), pepper (Liu et al 2022) and several other fruit crops (Wang et al., 2021). Likewise, fruit cracking is a major cause of fruit loss in pomegranate (Singh et al., 2020; Wang et al., 2022). However, nothing was mentioned about that in the manuscript, therefore, please add a paragraph about pomegranate cracking and expansins and discuss that.

Response 13: I have added it in the first paragraph of discussion.

Since cell wall enlargement played a key role in plant morphogenesis, the mechanism of cell wall extension had been a focus of investigation. During turgor-mediated growth, cell wall stress relaxation occurred, which involved cell wall-loosening factors such as EXPs. Expansins might offer a mechanism for cell wall elongation by interfering with the binding of microfibrils to the cell wall. The expansin gene family has been thoroughly discovered in numerous plants. Systematic analyses have previously been conducted to identify and characterize expansin families in several models, crop and fruit plant species, such as tomatos, pepper, litchi, cherry and watermelon. Subsequently, researches showed that a close relationship was established between expansins and fruit cracking. Fruit cracking tremendously damages the appearance of fruit, easily leads to pathogen invasion, greatly reduces the marketability and causes immense economic losses. Currently, pomegranate fresh fruit, seedlings and processing industry have a great promising future, while fruit cracking is a major cause of fruit loss in pomegranate. Thus, it is considerable and urgent to explore the roles expansins play in pomegranate.

Point 14: These are repeated results not conclusion, please remove and add your own conclusions, you presented tremendous data!

Response 14: I have revised the conclusions.

This work was the first comprehensive genome-wide analysis of the expansin gene family in pomegranate. We identified 33 PgEXPs from the pomegranate 'Taishanhong' genome. Based on a phylogenetic tree of pomegranate, Arabidopsis thalianaVitis vinifera, Actinidia chinensis and Ziziphus ziziphus, pomegranate genes were split into four subfamilies, EXPA (25 members), EXPB (5 members), EXLA (1 member) and EXLB (2 members). Members of subfamilies were highly conserved in motif and gene structure. Analysis of tissue-specific expression patterns of PgEXPs revealed that they may function differently in regulating organ/tissue morphology formation and development. Analysis of promoter cis-acting elements revealed that PgEXPs may respond to development and stress. qRT-PCR results played a role in exploring the function of PgEXP genes in pomegranate. The above analysis added to our knowledge of the expansin gene family in pomegranate and provided insights into the possible functional involvement of pomegranate expansin genes.

Point 15: It is recommended to incorporate more RECENT articles as only one fifth (11 out from 51) of  cited articles were published in the last five years.

Response 15: I have reviced them and incorporate more recent articles.

Reviewer 3 Report

The manuscript titled “Uncovering the Expansin Gene Family in Pomegranate: Ge- 2 nomic Identification and Expression Analysis” by Xu et al tries to characterize Expansin gene family in Pomegranates. The paper has potential but needs to improve future. I will request the authors to add the following issues- 

1.     Kindly use the updated version of MEGA software to generate the phylogenetic tree “The 103 expansin sequences from pomegranate (33), Arabidopsis thaliana (35), and Eu- 112 calyptus grandis (35) were aligned using the MAFFT. Based on the comparison results, the 113 phylogenetic tree was constructed by MEGA 7.0 software”.

2.     Why only two species were used for the tree construction “To investigate the phylogenetic relationships, a phylogenetic tree was constructed 175 using 103 protein sequences of EXP genes from pomegranate, Arabidopsis thaliana, and 176 Eucalyptus grandis.”? I would rather suggest adding the available sequence as mentioned in your introduction section “Arabidopsis [14], grape [15], apple [16], 52 cotton [17], kiwi [18], and cannabis”. Then the tree might take a better shape and more informative.

3.     There is no point in adding “The String protein interaction database was used to predict the co-expression of 33 273 PgEXP (Figure 5A) proteins (Arabidopsis thaliana was chosen as the model species and the 274 AtEXP with the highest similarity was selected).” now-a-days without experimental validation. Moreover, this is not a new gene family where prediction might have some significance.

4.     There is a lot of bar diagrams in figure 7. It looks repetitive. Moreover, represent the data as fold change in expression rather than relative expression in case of stress treatment, which will give better resolution.

5. Kindly be more specific in terms of plant variety, growing condition, age of the pants and other factors for qRT-PCR.

Author Response

Response to Reviewer 3 Comments

Point 1:  Kindly use the updated version of MEGA software to generate the phylogenetic tree “The 103 expansin sequences from pomegranate (33), Arabidopsis thaliana (35), and Eu- 112 calyptus grandis (35) were aligned using the MAFFT. Based on the comparison results, the 103 phylogenetic tree was constructed by MEGA 7.0 software”.

Response 1: I have updated version of MEGA software to align the selected protein sequences.

Point 2:  Why only two species were used for the tree construction “To investigate the phylogenetic relationships, a phylogenetic tree was constructed 175 using 103 protein sequences of EXP genes from pomegranate, Arabidopsis thaliana, and 176 Eucalyptus grandis.”? I would rather suggest adding the available sequence as mentioned in your introduction section “Arabidopsis [14], grape [15], apple [16], 52 cotton [17], kiwi [18], and cannabis”. Then the tree might take a better shape and more informative.

Response 2: The selected amino acid sequences of EXPs from pomegranate, Arabidopsis thalianaVitis vinifera, Actinidia chinensis, Ziziphus ziziphus were aligned by MUSCLE in MEGA 11 software with default settings, and the poorly aligned sequences were trimmed using the trimAL V1.4.1. The edited alignments were then analyzed using Maximum likelihood (ML) in IQTree2 V2.1.3 under the best-fitting model with 1000 bootstrap replicates.

Point 3: There is no point in adding “The String protein interaction database was used to predict the co-expression of 33 PgEXP (Figure 5A) proteins (Arabidopsis thaliana was chosen as the model species and the AtEXP with the highest similarity was selected).” now-a-days without experimental validation. Moreover, this is not a new gene family where prediction might have some significance.

Response 3: Likewise, fruit cracking is a major cause of fruit loss in pomegranate, but the function of the potential expansin genes were not yet clear. To date, a conserved number of expansins gene family had been identified and functionally characterised in both model plants such as Arabidopsis. However, no data set of expansins is available for pomegranate. Thus, exploring the functionally similar genes from AtEXPs may lay a foundation for the subsequent excavation of gene functions in pomegranate.

Point 4: There is a lot of bar diagrams in figure 7. It looks repetitive. Moreover, represent the data as fold change in expression rather than relative expression in case of stress treatment, which will give better resolution.

Response 4: The vertical bars represented the expression patterns of 24 PgEXPs in pomegranate peel at 6 developmental periods obtained by qRT-PCR analysis. Using a bar graph gave a more visual sense of gene expression, and I consulted the following articles, which have been published in recent years, when I analyzed this section.

Wang Y, Guo L, Zhao Y, Zhao X, Yuan Z. Systematic Analysis and Expression Profiles of the 4-Coumarate: CoA Ligase (4CL) Gene Family in Pomegranate (Punica granatum L.). International Journal of Molecular Sciences. 2022; 23(7):3509.

Zhang X, Wang S, Ren Y, Gan C, Li B, Fan Y, Zhao X, Yuan Z. Identification, Analysis and Gene Cloning of the SWEET Gene Family Provide Insights into Sugar Transport in Pomegranate (Punica granatum). International Journal of Molecular Sciences. 2022; 23(5):2471.

Hou, L., Zhang, Z., Dou, S. et al. Genome-wide identification, characterization, and expression analysis of the expansin gene family in Chinese jujube (Ziziphus jujuba Mill.). Planta 249, 815–829 (2019).

Zhang, S., Xu, R., Gao, Z. et al. A genome-wide analysis of the expansin genes in Malus × Domestica . Mol Genet Genomics. 289, 225–236 (2014).

Point 5: Kindly be more specific in terms of plant variety, growing condition, age of the pants and other factors for qRT-PCR.

Response 5: I have added some details in 2.1 section. The pomegranate variety for testing was ‘Daqingpitian’, and the sampling location was the Chinese Pomegranate Expo Park (34°77′N,117°48′E) with sloppy management level. We selected three healthy, disease-free and uniformly growing adult fruit-bearing trees, and took a mixed sampling method to collect samples. A total of six different developmental periods (the dates were 25 August, 3 September, 12 September, 21 September, 30 September, 9 October, designated as P1~P6, respectively) fruit samples were collected, finally.

Reviewer 4 Report

The manuscript deals with the genomic identification and expression analysis of expansin gene family in pomegranate. As the tree itself is an economically important species for its fruit peel and juice extracts that are rich in antioxidants, such as polyphenols, the analysis of this gene family that is known to be involved in plant development and fruit ripening is of interest to both pomegranate breeders and producers. In this context the manuscript brings a valuable information for the readers of Horticulturae journal.

While appreciating the overall value of the manuscript I have the following remarks and comments on its contents:

1.      Authors declare in the Conclusions that they have ”… found 33 PgEXPs from the pomegranate 'Taishanhong' genome, all of which contained two conserved domains (Pollen alleng_1 and DPBB_1)…” – Lines 410-411. This is not a surprising outcome as the approach used by the authors for the initial PgEXPs search in the pomegranate genomic sequences was based on looking for exactly these two conserved motifs. Therefore, putting such text in the conclusions is misleading and it needs to be removed.

2.      Similarly, the conclusion that the “Sequence comparison and structural characterization revealed that genes in the same group were structurally and functionally substantially conserved” is not surprising as the initial selection was for the two highly conserved domains. However, the way this conclusion is formulated gives the impression that this is valid for all the expansin genes in pomegranate, which might not prove correct if other initial search criteria are to be used. Therefore, this part of the conclusions also needs to be modified to correctly reflect the limitations of the initial experimental design.

3.      In the context of the previous comment, authors should further add in the discussion what other approaches to initial screening for EXPs in the genomes of various plant species were used and what was their reason to not apply them.

4.      In addition to these more substantial remarks, there are also some more technical issues to be addressed.

a.      While the volumes of both primer pairs and cDNA solutions are listed in the text (Lines 147-148) their quantities remain unknown as no concentration of these solutions is given. This needs to be corrected to make sure the results are reproducible by other authors.

b.      There is a strange comma, followed by double space in Line 69 (after the word Afghanistan), which seem to be a result of some incomplete text editing and need to be corrected.

c.      English needs moderate editing as in many cases improper usage can be found (i.e. “… compare sequences with all two conserved domains …”, Line 99).

Author Response

Response to Reviewer 4 Comments

Point 1:  Authors declare in the Conclusions that they have ”… found 33 PgEXPs from the pomegranate 'Taishanhong' genome, all of which contained two conserved domains (Pollen alleng_1 and DPBB_1)…” – Lines 410-411. This is not a surprising outcome as the approach used by the authors for the initial PgEXPs search in the pomegranate genomic sequences was based on looking for exactly these two conserved motifs. Therefore, putting such text in the conclusions is misleading and it needs to be removed.z

Response 1: I have deleted it.

Point 2:  Similarly, the conclusion that the “Sequence comparison and structural characterization revealed that genes in the same group were structurally and functionally substantially conserved” is not surprising as the initial selection was for the two highly conserved domains. However, the way this conclusion is formulated gives the impression that this is valid for all the expansin genes in pomegranate, which might not prove correct if other initial search criteria are to be used. Therefore, this part of the conclusions also needs to be modified to correctly reflect the limitations of the initial experimental design.

Response 2: I have revised the conclusions.

This work was the first comprehensive genome-wide analysis of the expansin gene family in pomegranate. We identified 33 PgEXPs from the pomegranate 'Taishanhong' genome. Based on a phylogenetic tree of pomegranate, Arabidopsis thalianaVitis vinifera, Actinidia chinensis and Ziziphus ziziphus, pomegranate genes were split into four subfamilies, EXPA (25 members), EXPB (5 members), EXLA (1 member) and EXLB (2 members). Members of subfamilies were highly conserved in motif and gene structure. Analysis of tissue-specific expression patterns of PgEXPs revealed that they may function differently in regulating organ/tissue morphology formation and development. Analysis of promoter cis-acting elements revealed that PgEXPs may respond to development and stress. qRT-PCR results played a role in exploring the function of PgEXP genes in pomegranate. The above analysis added to our knowledge of the expansin gene family in pomegranate and provided insights into the possible functional involvement of pomegranate expansin genes.

Point 3:  In the context of the previous comment, authors should further add in the discussion what other approaches to initial screening for EXPs in the genomes of various plant species were used and what was their reason to not apply them.

Response 3: To identify the members of the expansin gene family, two different approaches were utilised, but less difference in them.The first method: Arabidopsis expansin gene sequences were used as query sequences to perform blast searches against the proteome and genome files. Stand-alone versions of BLASTP and TBLASTN were used and the e-value cut-off was set to 1e−003. All protein sequences were examined using PFAM with 1.0 E-value cut-off and SMART with the default cut-off parameters. The second method was that analysed the domains of all peptide sequences using Hidden Markov Models (HMMs) with Pfam. Then, we can use ClustalX to verify candidate expansins. The following articles were consulted:

Wang Y, Guo L, Zhao Y, Zhao X, Yuan Z. Systematic Analysis and Expression Profiles of the 4-Coumarate: CoA Ligase (4CL) Gene Family in Pomegranate (Punica granatum L.). International Journal of Molecular Sciences. 2022; 23(7):3509.

Zhang X, Wang S, Ren Y, Gan C, Li B, Fan Y, Zhao X, Yuan Z. Identification, Analysis and Gene Cloning of the SWEET Gene Family Provide Insights into Sugar Transport in Pomegranate (Punica granatum). International Journal of Molecular Sciences. 2022; 23(5):2471.

Hou, L., Zhang, Z., Dou, S. et al. Genome-wide identification, characterization, and expression analysis of the expansin gene family in Chinese jujube (Ziziphus jujuba Mill.). Planta 249, 815–829 (2019).

Zhang, S., Xu, R., Gao, Z. et al. A genome-wide analysis of the expansin genes in Malus × Domestica . Mol Genet Genomics. 289, 225–236 (2014).

Point 4: In addition to these more substantial remarks, there are also some more technical issues to be addressed.

  1. While the volumes of both primer pairs and cDNA solutions are listed in the text (Lines 147-148) their quantities remain unknown as no concentration of these solutions is given. This needs to be corrected to make sure the results are reproducible by other authors.
  2. There is a strange comma, followed by double space in Line 69 (after the word Afghanistan), which seem to be a result of some incomplete text editing and need to be corrected.
  3. English needs moderate editing as in many cases improper usage can be found (i.e. “… compare sequences with all two conserved domains …”, Line 99).

Response 4:

  1. I have added them in the article.The PCR reaction system was 20 μL, including 10 μL Taq Pro Universal SYBR qPCR Master Mix (Vazyme, Nanjing), 0.4 μL upstream and downstream primers, 1 μL cDNA template (the concentration was around 200 ng/μL) and 8.2 μL ddH2
  2. I have revised it.
  3. Replace “Then we used HMMsearch to compare sequences with all two conserved domains (E-value≤10-10) in the whole genome of 'Taishanhong' pomegranate to get the amino acid sequences of the originally screened pomegranate expansins.” with “Then, using HMMsearch, we compared sequences with two domains (E-value≤10-10) in the pomegranate protein database to get the amino acid sequences of the originally screened pomegranate EXPs.”

Reviewer 5 Report

Figures need to be updated with higher resolution to be readable in order to understand the results.

Author Response

Response to Reviewer 5 Comments

Point 1: Figures need to be updated with higher resolution to be readable in order to understand the results.

Response 1: I have replaced all figures with higher resolution versions, and put them in supplementary materias for viewing.

Round 2

Reviewer 2 Report

25.4.2023

The revised manuscript entitled "Uncovering the expansin gene family in pomegranate: Genomic identification and expression analysis" was reviewed again.

I would like to thank the authors for taking into consideration most raised comments and suggestions. The subsisted revised manuscript was dramatically improved. There is still one major issue about using SD in qPCR. Although this is a common error published in 100s of articles, however, it is my duty to spread the correct method that can be used in qPCR data. fold change in qPCR data are relative data and raised as power of the figure 2. One can for examples utilize the SE to calculate the 95% confidence interval (95% CI, z score = 1.96), where C.I. = Mean ΔΔCt ± (z x SE). Nonetheless, I recommend the publication of the revised manuscript in "Horticulturae".

Author Response

Thank you again for this comment, it makes me realize the inadequacy of the knowledge I currently have. However, the time given for this revision process was only two days, this was not enough for me to fully understand the approach you provided and to complete the article revision. Therefore, I hope that you will understand that I apply it to my later experiments. Best wishes.

Reviewer 3 Report

The authors have addressed all the questions satisfactorily raised by me.

Author Response

I have checked my article sentence by sentence. Thank you for your previous comments, they make my article better. 

Reviewer 4 Report

The manuscript is much improved and can be published after editing English for clarity

Author Response

(The authors gave the same response as above.)
